# Cyanogenesis, a Plant Defence Strategy against Herbivores

**DOI:** 10.3390/ijms24086982

**Published:** 2023-04-10

**Authors:** Marta Boter, Isabel Diaz

**Affiliations:** 1Centro de Biotecnología y Genómica de Plantas (CBGP), Universidad Politécnica de Madrid (UPM)-Instituto Nacional de Investigación y Tecnología Agraria y Alimentaria (INIA/CSIC), Campus de Montegancedo, 20223 Madrid, Spain; 2Departamento de Biotecnología-Biología Vegetal, Escuela Técnica Superior de Ingeniería Agronómica, Alimentaria y de Biosistemas, Universidad Politécnica de Madrid (UPM), 28040 Madrid, Spain

**Keywords:** cyanide, cyanogenesis, cyanohydrins, cyanide detoxification, cyanogenic glucosides, defence strategy, phytophagous insects and mites

## Abstract

Plants and phytophagous arthropods have coevolved in a long battle for survival. Plants respond to phytophagous feeders by producing a battery of antiherbivore chemical defences, while herbivores try to adapt to their hosts by attenuating the toxic effect of the defence compounds. Cyanogenic glucosides are a widespread group of defence chemicals that come from cyanogenic plants. Among the non-cyanogenic ones, the Brassicaceae family has evolved an alternative cyanogenic pathway to produce cyanohydrin as a way to expand defences. When a plant tissue is disrupted by an herbivore attack, cyanogenic substrates are brought into contact with degrading enzymes that cause the release of toxic hydrogen cyanide and derived carbonyl compounds. In this review, we focus our attention on the plant metabolic pathways linked to cyanogenesis to generate cyanide. It also highlights the role of cyanogenesis as a key defence mechanism of plants to fight against herbivore arthropods, and we discuss the potential of cyanogenesis-derived molecules as alternative strategies for pest control.

## 1. Introduction

Plants have evolved a combination of physical and chemical strategies to precisely respond to phytophagous feeders. Specific information derived from damaged tissue, feeding guild and egg deposition is perceived by plant hosts and, together with their ability to recognize herbivore associated molecular patterns (HAMPS), allows plants to react accordingly in a highly specific manner [1,2,3]. A concomitant evolutionary process has taken place by phytophagous arthropods who have used plants as a source of nutrition over millions of years of coexistence. They have adapted their behaviour habits and physiology to their hosts, developing multiple tactics to overcome plant defences [4,5]. The outcome of this plant–pest interplay is a trade-off to ensure both adversaries survival, a balance that can be upset when environmental conditions favour one of them.

From the plant side, a huge battery of plant defences with a variety of chemical structures, biosynthetic pathways, and different modes of action have evolved under the selection pressure imposed by harmful interacting organisms. In this regard, the induction of secondary metabolites with defence properties is crucial, and among them, nitrogen-containing secondary compounds work very effectively to combat phytophagous insects and mites. In particular, there is a group of metabolites linked to cyanogenesis that have important protective roles. Plant cyanogenesis is defined as the release of hydrogen cyanide (HCN) as a consequence of tissue disruption triggered by herbivores or by physical injury [6]. The core of this specific physiological process of defence is based on the enzymatic hydrolysis of cyanohydrins to generate carbonyl compounds with free toxic cyanide (Figure 1). In the case of cyanogenic plants that contain cyanogenic glucosides (CNglcs), a previous reaction of de-glycosylation is required to separate the sugar moiety from CNglcs and liberate cyanohydrins. Thus, the cyanogenesis process in cyanogenic plants needs a typical two-component defence system to synthesize cyanides as chemical defence compounds [7]. Chemically, cyanohydrins or hydroxynitriles are organic compounds containing a cyanide and a hydroxyl group attached to the same carbon atom [8]. Cyanohydrins are metabolic intermediates derived from a set of molecular reactions rising from initial compounds, mainly amino acids, to finally be transformed into HCN and other toxic secondary metabolites [9].

The toxicity of HCN is due to its affinity for the ferric heme form of cytochrome *c* oxidase, the final enzyme in the respiratory chain. HCN acts as a potent respiratory poison since the formation of the cytochrome oxidase–CN complex blocks mitochondrial electron transfer and provokes cytotoxic hypoxia and death [10]. Other metabolites produced as a result of cyanohydrin breakdown along with HCN, such as benzaldehyde or benzoyl cyanide, are also toxic to pests. In addition, the benzoyl cyanide, has been associated with the production of hydrogen peroxide (H_2_O_2_) and is considered a signalling or defensive compound depending on its concentration [11]. While elevated levels of H_2_O_2_ induce cell death, moderate levels signal defence responses [12]. Cyanohydrins and derivatives such as mandelonitrile or ketone-cyanohydrin were also found to be hazardous substances against pests [13]. Thus, cyanogenesis is an excellent defence strategy for plants since it generates a set of toxic metabolites directly targeted to interfere with the phytophagous physiology. 

Those plant species in which cyanogenesis take place are termed cyanogenic plants. Nevertheless, HCN is a volatile molecule not only released from cyanogenic precursors such as CNglcs or cyanolipids, but also derived from the 4-hydroxy-indole-3-carbonylnitrile (4-OH-ICN) pathway described in non-cyanogenic Brassicaceae species [9,14]. Additionally, low levels of HCN are formed as a coproduct of ethylene biosynthesis or by camalexin degradation [15,16,17]. 

Therefore, facing challenges of secondary metabolites and their association with defences against herbivores is a goal, since pests are one of the major causes of crop yield losses. Accordingly, a wider understanding of a precise physiological process of defence such as cyanogenesis and its impact on phytophagous insects and mites could help to develop new strategies for pest control. This review compiles the current knowledge of cyanogenesis events linked to defences to herbivores, the complex network of interacting pathways, and the potential of cyanogenesis-derived molecules with defensive properties. It also includes some features related to cyanide detoxification.

## 2. Metabolic Pathways Linked to Cyanogenesis

### 2.1. Cyanogenic Glucosides (CNglcs): Biosynthesis and Catabolism

More than 3000 plant species, including economically important crops, are cyanogenic species, able to synthesize CNglcs and cyanolipids. Naturally occurring CNglcs have been identify in Angyosperms, especially in members of the Fabaceae, Poaceae, Rosaceae and Asteraceae families, plus in conifers and ferns [18]. CNglcs are nitrogen secondary metabolites containing an α-hydroxynitrile stabilized by a sugar, mainly a mono-glucoside such as glucose, and with less prevalence, di-glucosides, generally glucose plus arabinose. Their structure and a tissue distribution vary among cyanogenic plant species [9]. As indicated above, CNglcs are formed as inactive precursors and need to be enzymatically activated by the mentioned two-component defence system to generate appropriate chemically active compounds against herbivores in the right tissue and at the right time. Their synthesis starts with the conversion of aromatic (Phe, Tyr) or aliphatic (Val, Ile, Leu, Trp) amino acids into respective oximes catalysed by cytochromes P450 of the CYP79 family (CYP79D1/D2/A1). Some CNglcs are also derived from non-proteinogenic amino acids. In a second step, oximes are converted into α-hydoxynitriles by the action of the cytochromes P450 of the CYP71, CYP736, CYP706 and CYP83 families, to be then glycosylated by UDP-Glycosyl-transferases (UGT85B/K) yielding CNglcs [9]. In intact plant tissues, CNglcs are confined in vacuoles and are not toxic by themselves. After tissue disruption mediated by feeding herbivores, CNglcs are brought into contact with β-glucosidases that break the β-glucosidic bond, thereby releasing α-hydroxynitriles (cyanohydrins) with defensive properties [19]. Finally, cyanohydrins are hydrolysed either spontaneously or by hydroxynitrile lyases (HNL) (Figure 1), dissociating into HCN and the corresponding aldehyde or ketone [9,20]. Thus, HCN together with other hydrolytic products have a potential use due to their deterrent and toxic properties against the herbivores that cause cell disruption. A scheme of CNglc biosynthesis and their hydrolysis (cyanogenesis) upon tissue disruption is presented in Figure 2.

While CNglcs are confined in vacuoles, β-glucosidases are located in the apoplast bound to the cell wall of dicot species, and in cytoplasts or chloroplasts in monocots [6]. HNLs localize in the cytoplasm and plasma membranes [20]. The different compartmentalization of the CNgls and enzymes helps to avoid the overproduction of HCN and its toxicity in the plant. The separation of CNglcs and their hydrolytic enzymes also occurs at the tissue level. For example, the CNglc known as dhurrin identified in *Sorghum bicolor* is located in leaf epidermal layers, while two specific β-glucosidases (DH1 and DH2) called dhurrinases and the sorghum HNL have been detected in leaf mesophyll tissue [21]. Moreover, CNglcs may be accumulated either in vegetative (preferentially in leaves) and/or in reproductive tissues (flowers and seeds), but in general, younger tissues are more cyanogenic than older ones [9,22]. 

The functionality of the mentioned enzymes involved in CNglc biosynthesis was demonstrated by Bak et al. [23]. They converted non-cyanogenic tobacco and Arabidopsis plants into two cyanogenic species by the stable co-expression of *CYP79A1* and *CYP71E1* genes from sorghum. This couple of genes encoded two enzymes involved in the conversion of Tyr into p-hydroxymandelonitrile, a precursor of dhurrin. More recently, Lai et al. [24] combined the *CYP79D71* gene from lima bean (*Phaseolus lunatus*) with the *CYP736A2* and *UGT85K3* genes from *Lotus japonicus* to have the complete set of enzymes required for the synthesis of linamarin and lotaustralin, two CNglcs derived from Val and Ile, respectively. The simultaneous expression of the three genes in transiently transformed *Nicotiana benthamiana* resulted in the production of linamarin and lotaustralin in the infiltrated tobacco leaves. Likewise, cyanogenesis was also detected in *N. benthamiana* leaves transiently transformed with the *BGD3* glucosidase-encoding gene specifically expressed in flowers of *L. japonicus* when incubated with exogenous CNglcs prunasin, lotaustralin and linamarin, as substrates [25]. On the contrary, targeted mutagenesis of the *CYP79D1* gene via CRISPR/Cas9-mediated genome editing reduced the levels of linamarin and its evolved cyanide up to seven-fold in *Manihot esculenta* leaves [26]. Focusing on the last reaction involved in the cyanogenesis, the overexpression of the HNL under the control of a double 35S promoter in *M. esculenta* plants showed that the enzyme activity increased in transgenic leaves and specially in transgenic roots in comparison to wild type (WT) plants. The high activity correlated with a greater cyanide volatilization and a substantially reduced level of acetone cyanohydrin [27]. Similarly, transgenic *M. esculenta* plants carrying a patatin promoter (root-specific) driving the overexpression of the *HNL* gene exhibited lower linamarin content in roots than in control plants, and they had elevated protein and free amino acid levels as a consequence of enhanced linamarin metabolism [28]. These biotechnological approaches emerge as new alternative systems to fight against herbivores since they allow the control of the cyanide production as well as the generation of novel cyanogenic plants from non-cyanogenic ones. From an applied point of view, non-cyanogenic plants have been protected from herbivores by spraying formulates containing CNglcs [29].

Besides their defensive function, CNglcs have also been involved in plant growth and development and as osmo-protectants and scavengers or reactive oxygen species associated with responses to adverse environments [30,31]. Finally, CNglcs also play an important role in the transportation and turnover of nitrogen and glucose during specific plant developmental stages [31].

### 2.2. 4-Hydroxy-Indole-3-Carbonyl Nitrile (4-OH-ICN) Pathway

Although CNglcs are the most common cyanogenic compounds, derivatives from the 4-OH-ICN pathway exclusively found in the Brassicaceae family share with them the production of cyanohydrins, which can be then hydrolysed by the action of the HNL enzyme, releasing HCN. Both types of molecules are Trp-derived metabolites that also have in common the production of reactive oxime intermediates [9,14,20].

The 4-OH-ICN pathway has been described in the model species *Arabidopsis thaliana* as a rare metabolic *reinvention* leading to an alternative cyanogenic indole carbonyl nitrile (ICN) to expand plant defences [14,32]. As is shown in Figure 3, in the initial step of this pathway, two redundant P450 monooxygenases, CYP79B2 and CYP79B3, convert tryptophan into indole-3-acetaldoxime (IAOx), which acts then as the substrate taken by CYP71A12 to produce indole cyanohydrin. At this point, a flavin-dependent oxidoreductase termed FOX1 catalyses the conversion of IAOx to indole-carbonylnitrile (ICN), an intermediate that can be transformed into 4-OH-ICN through the CYP82C2 enzyme. Additionally, 4-OH-ICN is the base for downstream cyanohydrin metabolite production. Rajniak et al. [14] reconstituted the biosynthesis of 4-OH-ICN in *N. benthamiana* by the agroinfiltration of the four mentioned genes (*CYP79B2, CYP71A12, FOX1* and *CYP82C2*) and observed the accumulation of 4-OH-ICN derivatives, in particular of 4-OH-indole-3-carbonyl methyl ester, validating the function of the pathway and the responsible enzymes. Thus, this pathway and their related gene-encoding enzymes can be used as a biotechnological alternative to induced cyanogenesis and to fight against phytophagous feeders.

Very recently, Arnaiz et al. [20] have characterized an HNL-encoding gene in Arabidopsis responsible for the reversible interconversion between cyanohydrins and derived carbonyl compounds with free cyanide. The authors found that the *HNL* gene was highly induced in response to spider mite *Tetranychus urticae,* infestation. Loss- and gain-of-function of this gene in Arabidopsis plants showed that the mandelonitrile cyanohydrin had accumulated at higher levels in mutant lines than in WT plants, while it was significantly reduced in AtHNL overexpressing lines. In this non-cyanogenic model species, HNL is clearly associated with the 4-OH-ICN pathway since this pathway supplies the substrate for the HNL enzyme to finally release HCN for plant protection [14] (Figure 3). The production of cyanohydrins derived from the 4-OH-ICN pathway together with the HNL action are the requirements needed to complete an alternative cyanogenic process and to expand plant defences.

### 2.3. Crosstalk of Indole Metabolic Related Pathways and Cyanogenesis Involved in Defence to Herbivores

Some authors have reported the extensive landscape of variations in amino-acid-derived oximes linked to different plant biosynthetic pathways, which are highly redirected to a secondary metabolism specialized in the defence against biotic stresses [14,24,32,33]. In the context of cyanogenic events, the Trp-derived oxime pathway is of crucial interest to account on containing convergence and bifurcation points among different metabolic routes associated with defences. Trp and its conversion into IAOx are the common starting point within this complex network of enzymatic pathways. Although oximes also have many other functions in plants [33], in relation to defence, IAOx acts as first intermediates for the biosynthesis of CNglcs in cyanogenic species, as well as for the synthesis of 4-OH-ICN derivatives, camalexin and indole glucosinolates (IGs) within Brasicaceae. The route followed by IAOx will be mainly determined by the hosts and feeder species (Figure 3). 

Camalexin is an indole alkaloid derived from Trp and induced in response to pest and pathogen attack. In Arabidopsis, IAOx metabolizing steps in camalexin biosynthesis involves additional cytochrome P450 reactions after the conversion of Trp into oximes. First, the transformation of IAOx into indole-3-acetonitrile (IAN) is catalysed by CYP71A12 and its close homologue, CYP71A13. Then, the formation of camalexin takes place through the action of the CYP71B15 monooxygenase, also known as PHYTOALEXIN DEFICIENT 3 (PAD3) [34]. The toxic properties of camalexin are not completely elucidated but seem to be linked to a direct effect on the membrane integrity, among other potential activities [35]. Camalexin’s defensive effects have been shown in aphids fed the Arabidopsis *pad3* and *cyp79b2/cyp79b3* mutants defective in its production. These aphids were more successful at feeding and had higher fecundity rates than those fed on WT plants [36]. 

Alternatively, the enzyme CYP83B1 channels oximes to the biosynthetic pathway for IG production. These nitrogen- and sulfur-containing metabolites are key defence compounds. IGs show very limited biological activity and, to become active forms to combat pests, require an additional two-component defence system, as CNglcs do, mediated in this case by myrosinases [7]. The importance of IGs for defence against herbivores is supported by the results of insect feeding assays on *A. thaliana* mutant lines, which make plants more susceptible to be attacked, mainly by specialist insects and spider mites. In this scenario, it is important to highlight a recent publication by Widemann et al. [37], which demonstrated that three IG metabolites—I3M, 1MO-I3M, and 4MO-I3M IG—were necessary and sufficient to protect Arabidopsis plants against the spider mite *T. urticae.*

The last point of convergence among different defensive pathways, all derived from Trp, corresponds to the final enzymatic reaction for HCN release, a key step for cyanogenesis (Figure 3). This catabolic reaction mediated by HNL is the crucial point to produce toxic secondary metabolites with defence properties. Curiously, three different metabolic pathways related to defence start and end with common compounds and search two catalysing enzymes; however, in between, a huge battery of molecules are generated. Further knowledge about the function of these secondary metabolites may shed light on the advantages of having points of metabolic convergence. 

## 3. Cyanohydrins, Cyanogenesis and the Control of Phytophagous Arthropods (Insects and Mites)

Many studies have experimentally shown the potential of cyanogenic plants in the control of arthropod pests, including hemipteran, coleopteran, orthopteran, and homopteran insects (Table 1). In general, insect damage negatively correlates with cyanogenic and phenolic compounds [30,38,39,40,41,42]. Some works using insect preference assays allowing feeders to choose among plants with different CNglc or cyanohydrin content have demonstrated that insects chose genotypes deficient in cyanide release as a source of nutrients or as the place to lay their eggs [43,44]. In addition, bioassays based on improved synthetic diets have been used to identify and test bioactive substances against pests. This experimental method has allowed the identification of dhurrin isolated from sorghum leaves as one of the major deterrents against the aphid *Schizaphis graminum* [45]. Hay-Roe et al. [46] also found that the fall armyworm *Spodoptera frugiperda* exhibited high mortality when fed an artificial diet supplemented with cyanide. Then, Mora et al. [47] developed a multilayer grain coating system in wheat consisting of biodegradable polylactic acid with individual layers containing the CNglc amygdalin or β-glucosidase isolated from bitter almond. When the layers were ruptured by the coleopteran *Tenebrio molitor*, *Rhizopertha dominica*, and *Plodia interpunctella*, their reproduction and the grain consumption rate were significantly reduced. This artificial application system could be expanded to many crops and had a clear potential to control storage pests. 

Other reports have employed mutant lines for genes encoding biosynthetic or catabolic enzymes involved in cyanohydrin pathways to check pest feeding responses. This is the case for sorghum mutants for *CYP79A1* and *dhr2* genes, which catalyse the first and last steps of dhurrin production, respectively. Data revealed that both mutants were selected by *S. frugiperda* larvae in a settling preference experiment over their isogenic control plants [43,48]. Transgenic approaches have also been used to express the complete pathway for dhurrin synthesis from sorghum in *A. thaliana*, resulting in protection against the flea beetle *Phyllotreta nemorum* [49]. The accumulation of dhurrin in Arabidopsis did not affect the plant’s physiology or growth, and it conferred resistance to this coleopteran, which is considered a common herbivore of crucifers. Altogether, the mentioned examples build a strong foundation stablishing that cyanogenesis events in cyanogenic plants are closely associated with plant defence mechanisms for protection against herbivores.

In contrast to the high amount of data demonstrating the role of cyanogenic plants in the control of arthropod pests (Table 1), as far as we know, there is just one publication validating the alternative cyanogenic pathway to expand plant defences against pests in a non-cyanogenic plant such as Arabidopsis. Arnaiz et al. [20] demonstrated the up-regulation of the At*HNL* gene in response to a spider mite infestation as well as the induction of the expression of the *FOX1* gene encoding the FAD-linked oxidoreductase 1 that catalyses the conversion of ICNs into 4-OH-ICN (Figure 3). These data indicated that this alternative pathway was triggered in Arabidopsis by mites. Moreover, a reduction in the leaf damage determined in the *AtHNL* overexpressing lines reflected the mite’s reduced ability to feed on leaves, which consequently limited mite fecundity. On the other hand, Arabidopsis *hnl* mutant lines were more susceptible to mite attack and accumulated higher numbers of eggs than WT-infested plants as a result. This set of information opened the door to the potential use of HNLs as a common convergence point of defence pathways to release cyanide (Figure 3).

Based on the defensive effects of cyanogenesis precursors and the derived compounds, the insecticidal activity of some cyanohydrins and cyanohydrin esters have been evaluated in topical applications to some insects, and this has shown that cyanohydrin bioactivity was similar to other proven fumigants [50]. Thus, fumigation with optimized preparations of natural and synthetic cyanohydrins to stored product were tested, demonstrating their valuable and potential application for plant pest control [13]. However, further research to determine their efficacy and economic feasibility is required.

## 4. Herbivore Responses to Cyanogenic Metabolites

Plant feeders are always exposed to certain amounts of HCN, although it is expected that cyanogenic plants could release more cyanides than non-cyanogenic ones. Moreover, plant cyanogenesis is dependent on the amount of tissue damage, which is determined by the arthropod feeding mode. Chewing insects stand out from others for causing great tissue disruption and a rapid release of toxic metabolites by triggering the two-component defence system that brings together CNglcs and glucosidase enzymes. In contrast, sucking species produce less cell damage and potentially minor plant responses [51,52]. Phloem-feeding insects, such as aphids, introduce their long sucking stylet to the feeding site via apoplast without disturbing cells to elude defences. Spider mites use open leaf stomata for stylet penetration, or alternatively their stylet penetrates between the epidermis pavement cells, reaching the mesophyll layer without damaging surrounding cells [53]. In addition, herbivores have evolved a battery of strategies and habits to prevent cyanogenesis. While generalist species try to feed in non-cyanogenic hosts, specialists can more easily metabolize toxic compounds or phosphorylate and glycosylate them to avoid cyanide production. Other insects maintain CNglcs intact to be secreted or can even sequester them from plants to be used as their own weapons against predators [9,54,55]. 

In addition, many organisms, including plants and arthropods, have developed detoxification mechanisms to keep cyanide below lethal levels. Toxic cyanide can be mainly detoxified through the mitochondrial β-cyanoalanine synthase (CAS), and to lesser extent by cyanases and rhodaneses [20,56]. CASs are the most efficient enzymes and transform the HCN into β-cyanoalanine and hydrogen sulfide. β-cyanoalanine has also deterrent properties to herbivores since it is a strong neurotoxin, but it can be converted into Asn, Asp, and NH4 by nitrilases of the T4-class in some species [54,56]. Cyanases require the previous oxidation of HCN into cyanate catalyzed by cyanide monoxygenases, to be then metabolized into ammonia and carbon dioxides [57]. Rhodaneses convert HCN into thiocyanate and sulphite [58]. Some reports have corroborated the detoxification of HCN mediated by CAS and rhodaneses in phytophagous insects and mites. Thus, Stauber et al. [59] detected an increase in the content of β-cyanoalanine and thiocyanate, both sub-products of the respective catalysis, in the lepidopteran *Pieris rapae* after feeding on transgenic Arabidopsis plants containing dhurrin. Likewise, the delivering of dsRNA-*TuCAS* to mites allowed Arnaiz et al. [20,22] to demonstrate the effect of CAS detoxification on mite performance, shown by a significant reduction in the fecundity of treated mites after feeding on Arabidopsis *hnl* mutant lines and its isogenic control plants. Similarly, combining Arabidopsis genetics, chemical complementation and mite reverse genetics, Dixit et al. [60] showed that CAS was required for mite adaptation to Arabidopsis to counter cyanide toxicity. These examples highlight the pest’s ability to respond to toxic metabolites by activating resistant traits to use cyanogenic plants as hosts.

## 5. Conclusions and Future Perspectives

Plants produce a high diversity of secondary metabolites that have relevant functions in the protection against predators based on their toxicity and repellence to herbivores. The combined action of different groups of plant defensive metabolites against herbivores is essential to cover and combat the whole range of plant feeders, including generalists and specialists with different feeding modes, behaviours, and capabilities to develop resistances. Although cyanogenesis is widely spread in the plant kingdom, most of our knowledge related to cyanogenic events derives from studying a limited number of plants and phytophagous arthropods. However, it seems clear that plants have independently evolved metabolic pathways linked to cyanogenesis several times, with common convergence points and specific divergent branches. Moreover, plants have engaged in new key enzymatic activities and intermediates to finally expand defences and synthesize protective metabolites. In the cyanogenenic process, tissue disruption as a result of chewing or sucking by an herbivore is necessary to bring together CNglcs and β-glucosidases that catalyse the cleavage of the sugar moiety from the cyanogenic glycoside, resulting in cyanohydrin production. Cyanohydrins, either derived from CNglcs or from the 4-OH-ICN pathway in Brassicaeae, are then degraded by an α-hydroxynitrile lyase enzyme to produce HCN and an aldehyde or a ketone (Figure 4). This dual system provides a rapid defence response to herbivores that cause tissue damage. The effectiveness of cyanogenesis as an herbivore deterrent is not universal and usually depends on the cyanohydrin concentration, the herbivore species and the feeding mode. Moreover, natural and synthetic cyanohydrins have been proven as active compounds of fumigants with positive effectiveness for pest control. A wider understanding of the cyanogenesis process and its emerging alternative roles in plant physiology would open new options and strategies to improve pest control measures and agricultural production.

## Figures and Tables

**Figure 1 ijms-24-06982-f001:**
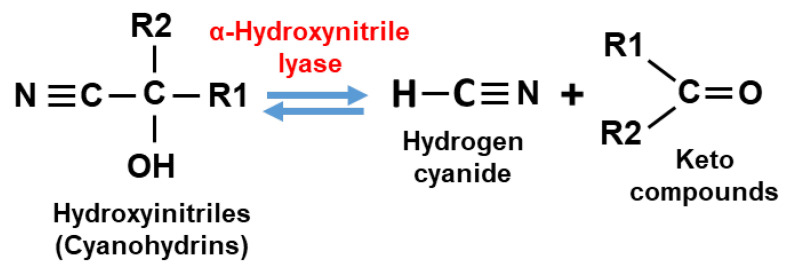
Enzymatic reaction for cyanohydrin and HCN interconversion catalysed by α-hydroxynitrile lyase.

**Figure 2 ijms-24-06982-f002:**
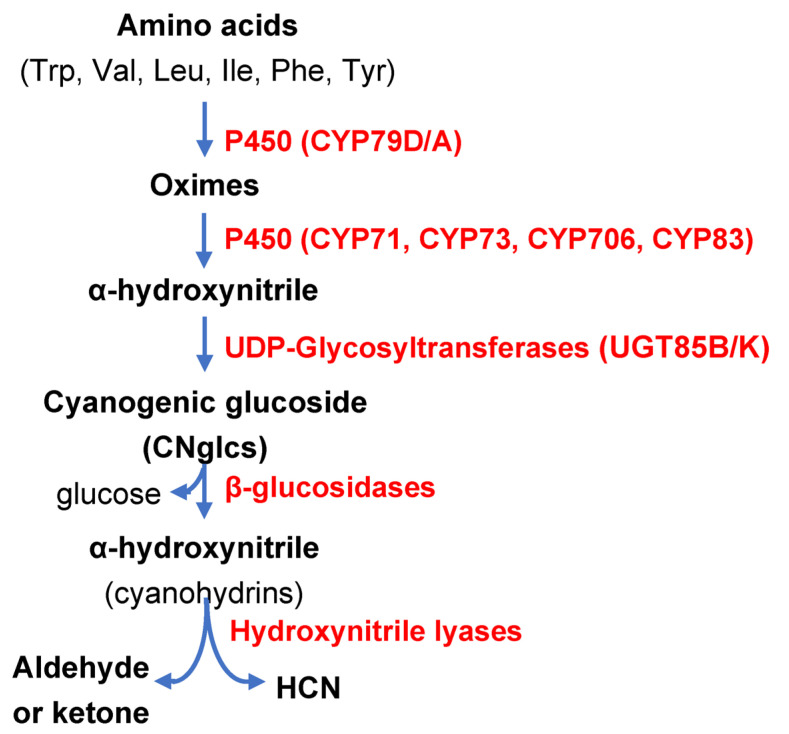
Scheme of biosynthesis and catabolism of Cyanogenic Glucosides (CNglcs). Enzymes are indicated in red colour.

**Figure 3 ijms-24-06982-f003:**
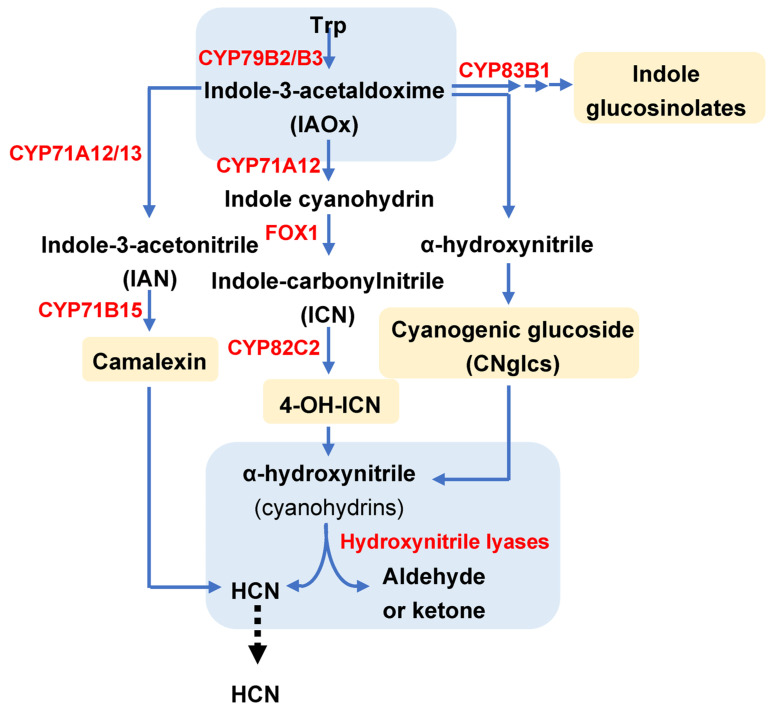
Scheme of different plant pathways involved in HCN production, dependent on the plant species. Enzymes are indicated in red. Yellow boxes indicate final products of each pathway, and blue boxes indicate common convergence points in the network of pathways. Adapted from Araniz et al. [20].

**Figure 4 ijms-24-06982-f004:**
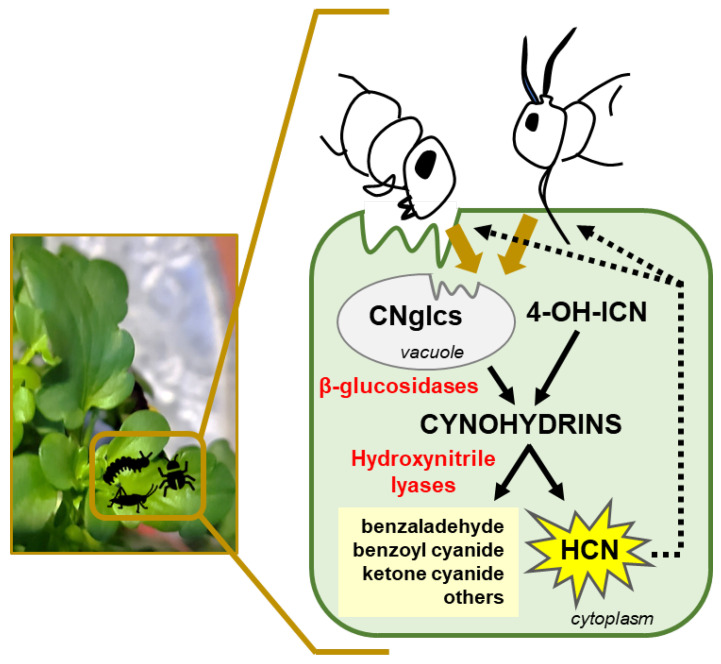
Cyanogenesis events in plant defence against pests. Insect and acari feeding on plants disrupt cells bringing in contact substrates (CNglcs: cyanogenic glucosides and 4-OH-ICN: 4-OH-Indolecarbonyl-nitrile derivatives) and enzymes (β-glucosidases and hydroxynitrile lyases). Consequently, HCN is released, and aldehyde- or ketone-compounds are produced to combat pests.

**Table 1 ijms-24-06982-t001:** Target phytophagous arthropods (insect or acari), plant or experimental assay and particular features (metabolite, enzyme or physiological characteristics) involved in cyanogenesis events.

Insect/Acari		Plant/Assay	Feature	Reference
Order	Species			
Lepidopteran	*Chilo partellus*	sorghum	cyanogenic	[41]
	*Spodoptera frugiperda*	synthetic diet	NaCN	[46]
	*Spodoptera frugiperda*	sorghum	cyanogenic	[43]
	*Plodia interpunctella*	wheat	amygdalin or β-glucosidase	[47]
	*Spodoptera littoralis*	lima bean	cyanogenic	[30]
	*Spodoptera frugiperda*	sorghum	cyanogenic	[48]
Hemipteran	*Cyrtomenus bergi*	cassava	cyanogenic	[40]
Coleopteran	*Phyllotreta nemorum*	Transgenic arabidopsis	dhurrin	[49]
	*Rizopertha dominica*	glass tube fumigant	cyanohydrins	[13]
	*Tribolium castaneum*	glass tube-fumigant	cyanohydrins	[13]
	*Sitophilus zeamias*	glass tube-fumigant	cyanohydrins	[13]
	*Oryzaephilus surinamensis*	glass tube-fumigant	cyanohydrins	[13]
	*Epilachna varivestis*	lima bean	cyanogenic	[39]
	*Tenebrio molitor*	wheat	amygdalin or β-glucosidase	[47]
	*Rizopertha dominica*	wheat	amygdalin or β-glucosidase	[47]
Orthopteran	*Locusta migratoria*	sorghum	cyanogenic	[42]
Homopteran	*Schizaphis graminum*	synthetic diet	dhurrin	[45]
Acari	*Tetranychus urticae*	transgenic arabidopsis	HNL	[20]

## Data Availability

Not applicable.

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
