# Peer review of "Cyanogenesis, a Plant Defence Strategy against Herbivores"

_ijms, 2023, doi:10.3390/ijms24086982_

Round 1

Reviewer 1 Report

This is very interesting manuscript focusing on the plant metabolic pathways linked to cyanogenesis to generate cyanide as well as on the role of cyanogenesis in plant defence mechanism. The high number of recently published references has been given in this review and such summarized knowledge is valuable for further research.

I think that minor changes must be made before its publishing.

Specific comments are listed below.

Page 1, line 35: “…environmental cues…” – I suggest use some other expression than cues

Page 1, lines 42-43: please add some reference

Page 2, lines 50-51: please correct the writing in the bold

Page 2, line 67: please add formula H2O2 after the “…hydrogen peroxide…”

Page 3, lines 100-101: “… catalysed by cytochromes P450 of the CYP79 family (CYP79D1/D2/A1)…” – the cytochromes P450 is not shown in the Fig.2 in the step Amino acid – Oximes, but in the next step Oximes - α-hydroxynitriles it is shown. Why?

Page 3, lines 103-104: UDP-Glycosyltransferases (UGT85B/K) is written differently in the text and in the figure 2. Please correct it

Page 3, 104: Please delete the parenthesis after the [8]

Page 3, 107-109: please check the explanation of the step from CNgls to α-hydroxynitriles and compere with the Fig.2. I think that some steps or explanation is missing?

Page 4, line 139: BDG3 or BGD3?

Page 4, line 143: Latin name was used for previously mentioned species; thus, I suggest use the Latin name and for cassava

Page 4, line 158: the role as osmo-protectants and ROS scavengers is not the main object of the reference 30. I suggest, add some references that better explained that roles

Page 4, line 161: please add a interspace between the 2.2 and 4-hydroxy….

Page 5, 171: different name was used for indole-3-acetaldoxime (IAOx) in the text and in the Fig.3 (indole-3-acetoxime (IAOx)) - please correct it

Page 5, 172: “…CYP71A12/13 to produce indole-cyanohydrin” but in the Fig. 3 this led to the Indole-3-acetonitrile (IAN) – please check and correct the explanation in the text and those showed in the Figure 3.

Page 5, line 177: CYP97B2 or CYP79B2?

Page 5, lines 167 and 189: model specie? – please correct

Page 6, line 220: I suggest add the full name of PAD3, firstly

Page 7, line 234: If you refer on some author and concrete publication I suggest use the same abbreviation for some enzymes as was shown in that pubication (e.g. 1-Methoxy-I3M (1MO-I3M)

Page 9, line 295: point t – please delete the double t

Page 9, line 303: put the point at the end of the sentence

Table 1: please check the reference 41 in the part Coleopteran, I think it is mistake

Table 1: please change Triboium castaneum into Tribolium castaneum

Table 1: reference 39 in the part Coleopteran has been written twice – please check this

References

Page 12, line 408-409: please check the journal abbreviation (I think it should be Curr. Opin. Plant Biol.)

Page 12, line 411: please check the journal abbreviation

Page 12, line 414: please correct the surname of the second author

Page 12, line 419: please check the journal data (I think it should be 2017, 104, 19)

Page 13, line 425: please write Arabidopsis in italic

Page 13, 429: please correct the journal pages

Page 13, line 430: please correct the name Garcia

Page 13, line 438: please check the spelling of the authors surname (Jorgensen, Moller)

Page 13, line 443: please check the spelling of the authors surname (Jorgensen, Sanchez-Perez, Moller), and write the year of publication in bold

Page 13, line 446: please correct the journal pages

Page 13, line 447-449: please correct the name of Georgie (something is missing), delete the year of publication in parenthesis (2017), and correct the year of publication (2018)

Page 13/14, lines 450-453: please check the spelling of the author surname (Moller), and delete the colon after the journal volume

Page 14, lines 454-455: please check and correct the spelling author surnames, one author is missing

Page 14, line 457: please check the spelling of the authors surname (Pismanova, Moller)

Page 14, line 474: Please correct the author surname (Moller into Møller), and journal abbreviation

Page 14, line 476: please check the spelling of the authors surname

Page 15, line 479: please check the spelling of the authors surname

Page 15, line 481: please check the spelling of the author surname (Bottcher); put the point after the journal name, add a comma between the journal volume and pages

Page 15, line 485: please check the spelling of the author surname (Clement)

Page 15, line 491-492: please check and correct this reference as well as other mentioned in the reference list

Author Response

Please, find attached the document with our responses.

Reviewer 2 Report

The topic is very interesting. I am pleasantly surprised by the topic addressed. The work as a whole is complex and fairly well organized, which shows the correct and detailed documentation of the authors regarding the chosen topic.

Figures and tables are well organized and correlated. Conclusions and future perspectives are short and concise. 

I would recommend a short linguistic check, there are some small errors.

Author Response

Thank you very much, we appreciate your comments. A native English-speaking colleague has corrected this final version of the manuscript.

We hope that this revised version of the manuscript can be finally accepted for publication.
